# Histone Deacetylases Function in the Control of Early Hematopoiesis and Erythropoiesis

**DOI:** 10.3390/ijms23179790

**Published:** 2022-08-29

**Authors:** Pascal Vong, Hakim Ouled-Haddou, Loïc Garçon

**Affiliations:** 1Université Picardie Jules Verne, HEMATIM UR4666, 80000 Amiens, France; 2Service d’Hématologie Biologique, Centre Hospitalier Universitaire, CEDEX 1, 80054 Amiens, France; 3Laboratoire de Génétique Constitutionnelle, Centre Hospitalier Universitaire, CEDEX 1, 80054 Amiens, France

**Keywords:** HDAC, hematopoiesis, erythropoiesis

## Abstract

Numerous studies have highlighted the role of post-translational modifications in the regulation of cell proliferation, differentiation and death. Among these modifications, acetylation modifies the physicochemical properties of proteins and modulates their activity, stability, localization and affinity for partner proteins. Through the deacetylation of a wide variety of functional and structural, nuclear and cytoplasmic proteins, histone deacetylases (HDACs) modulate important cellular processes, including hematopoiesis, during which different HDACs, by controlling gene expression or by regulating non-histone protein functions, act sequentially to provide a fine regulation of the differentiation process both in early hematopoietic stem cells and in more mature progenitors. Considering that HDAC inhibitors represent promising targets in cancer treatment, it is necessary to decipher the role of HDACs during hematopoiesis which could be impacted by these therapies. This review will highlight the main mechanisms by which HDACs control the hematopoietic stem cell fate, particularly in the erythroid lineage.

## 1. Introduction

The hypothesis that reversible histone acetylation may control transcriptional activity was proposed in the 1960s [1,2]. Subsequently, HATs and HDACs were identified. These enzymes are capable of covalently modifying the amino-ε side chain groups of the lysine residues of core histones, thereby not only impacting the regulation of gene expression, but also of non-histone proteins [3,4,5,6,7]. In humans, there are 18 HDACs involved in physiological functions and tumoral pathologies. Due to structural variations, HDACs differ in their enzymatic mechanism and do not possess the same substrate specificity. Furthermore, these proteins do not share the same tissue and intracellular localization. They can be categorized into four classes based on their sequence and tertiary structure homology to yeast HDACs (Table 1). Class I consists of the yeast deacetylase homologs RPD3: HDAC1, -2, -3, and -8 [8,9,10,11,12]. They are ubiquitous and localized primarily in the nucleus. Class II HDACs are related to the yeast HDA1 protein [13,14,15,16,17]. This class can be separated into two subgroups of enzymes that can shuttle between the cytoplasm and the nucleus: class IIa, which includes HDAC4, -5 and -7, and class IIb, which includes HDAC6 and -10 [18,19,20,21,22]. HDAC6 is the major cytoplasmic deacetylase in humans [18,23], and is notably characterized by the presence of two deacetylase domains and a C-terminal zinc finger motif. Class III includes sirtuins 1–7, whose catalytic activity depends on the coenzyme NAD+, to which they can bind, unlike other HDACs [6,24]. They are located in various cellular compartments, such as the nucleus, cytoplasm, and mitochondria [25]. Class IV is represented by a single member: HDAC 11. This protein is found in both the nucleus and the cytoplasm [26,27,28].

## 2. Role of HAT and HDAC in the Regulation of Gene Expression though Histone Acetylation

Since the discovery of the correlation between histone acetylation levels and gene transcriptional activity in 1964 by Allfrey et al. [1], core histones have become the best-characterized target proteins for acetylation and deacetylation processes. These processes play an important role in the regulation of transcription in eukaryotic cells [29,30,31]. The acetylation status of core histones is regulated by the antagonistic activities of HATs and HDACs. HATs add an acetyl group to the lysine residues of histone amino tails, whereas HDACs remove it [32]. Electrostatic interactions between the negatively charged phosphodiester backbone of DNA and the positively charged basic residues of the N-terminal tails of core histones maintain chromatin in a tightly packed structure. The positive charge carried by the lysine residue can be neutralized when acetylated and restored when deacetylated [33]. Thus, acetylation of histones by HAT produces chromatin loosening, which promotes the accessibility of DNA to transcriptional complexes [34,35]. Conversely, by deacetylating histones, HDACs restore positive charges and thus promote chromatin compaction into a conformation that is repressive in most cellular processes [33,36]. Tighter DNA coiling is indeed associated with the decreased accessibility of transcription factors to DNA, thereby reducing gene expression [37,38,39].

However, the relationship between HDACs and gene expression in yeast remains controversial. There is evidence that HDACs are highly present in transcriptionally active genes and may therefore also be involved in transcription [40,41]. Indeed, the yeast HDAC complexes RPD3 and RPD1 are involved in both transcriptional repression and activation [42,43]. In support of this hypothesis, recent studies revealed that HDAC inhibitors (HDACi) can repress the transcription of many genes [44,45,46]. The functional link between HATs (CBP, p300, PCAF, Tip60, MOF) and HDACs (HDAC1, HDAC2, HDAC3, HDAC6) has been studied by Wang et al. who propose a model according to which HATs and HDACs act in a collaborative mode. These two families of enzymes with antagonistic activities are recruited to active chromatin to promote gene expression and histone acetylation [47]. Indeed, genome-wide mapping of chromatin binding to HAT or HDAC showed that these enzymes localize to transcriptionally active regions with acetylated histones. The phosphorylated RNA Pol II commonly mediates their recruitment. Wang et al. suggest that HDACs play two major roles at the chromatin level. Its first function takes place at the level of activated genes and allows a chromatin reset by suppressing acetylation produced by HATs. This step is necessary to initiate the next cycle of transcription. The second function of HDACs occurs at the level of inactivated genes: inactive genes that are primed by histone H3K4 methylation undergo a dynamic cycle of acetylation and deacetylation by transient HAT/HDAC linkages. The low level of acetylation resulting from this dynamic process prevents RNA Pol II binding, which keeps gene promoters in an inactive state, but prepares them for future activation.

## 3. Biological Function of Acetylation/Deacetylation of Non-Histone Substrates

In addition to histones, several cytoplasmic and nuclear non-histone substrates are also targets of reversible acetylation [48]. Phylogenetic analyses have revealed that the HDAC family enzymes expressed in bacteria are known to lack histones. This suggests that acetylation and deacetylation of non-histone proteins are highly conserved biological processes, and that HDACs essentially deacetylated non-histone proteins at an early stage of evolution [49,50]. The transcription factor p53 is an example of a non-histone nuclear protein that can undergo reversible acetylation. Acetylation of p53 is associated with various biological effects, such as cell cycle arrest, DNA repair, senescence, or cell apoptosis. Many HATs, such as Tip60, MOF and p300/CBP, are implicated in p53 acetylation [51,52,53]. The HAT p300, by acetylating p53, produces a conformational change that increases its transcriptional activity [54]. Tip60 directly stimulates p53 apoptotic activity through its acetyltransferase activity at the lysine residue K120 [53]. Tip60 also protects p53 from its proteasomal degradation by binding to the Mdm2 antagonist of p53 [55,56]. Other transcription factors have also been identified as target substrates for acetylation and deacetylation processes, such as Ku70 [57], the RelA subunit of NF-κB [58], or the proto-oncogene protein, c-Myc [59].

Highlighting the importance of the acetylation/deacetylation of non-histone substrates in cell physiology, HDAC6, is remarkable because of its predominant cytoplasmic localization, which depends on the nuclear export sequences and the cytoplasmic retention sequence [60]. In the cytoplasmic compartment, HDAC6-mediated deacetylation regulates several cellular processes, such as clearance of misfolded and ubiquitinated proteins through the formation of aggresomes [61], autophagic maturation [62] and the induction of chaperone protein expression [63], or the cytoskeleton organization and stability strongly linked to the level of acetylation of α-Tubulin, one of HDAC6 best-characterized substrates. HDAC6-mediated α-Tubulin deacetylation at residue K40 is a marker of stable microtubules. This post-translation modification increases the resilience of microtubules, protects them from mechanical stresses [64] and facilitates the recruitment of proteins, such as kinesins [65,66], HSP90 [67], or enzymes involved in microtubule fragmentation, such as katanine [68]. Cortactine (CTTN), a monomeric cytoskeletal protein involved in the polymerization and remodeling of the actin cytoskeleton [69], is another well-identified HDAC6 target, and plays a crucial role during late megakaryopoiesis. Indeed, mature megakaryocytes (MKs) emit cytoplasmic extensions called pseudopodia, or proplatelets (PPT), which cross the wall of the medullary sinusoids and fragment to release platelets into the lumen of the sinusoids [70,71,72]. PPT branching and MK fragmentation are regulated by microtubules and actin polymerization [73,74]. In human MKs, pharmacological or genetic invalidation of HDAC6 results in the hyperacetylation of CTTN in vitro, reducing its ability to interact with F-actin, which subsequently impairs its polymerization into actin filaments and thus the terminal differentiation of MKs [75].

## 4. HDACs in Hematopoiesis

Many studies using pan-HDAC inhibitors have shown that HDACs have a broad spectrum of functions in human and murine hematopoietic stem cells and in more committed cells, particularly in the erythroid lineage. Main functions of HDACs during hematopoiesis are summarized Table 2.

### 4.1. HDAC Role in the Maintenance of Hematopoietic Stem Cell (HSC)

HDAC1 and HDAC2 are class I HDACs that play an essential role in the regulation of HSC homeostasis in conditional knockout murine models. Simultaneous deletion of HDAC1 and HDAC2 results in the loss of HSC and consequently bone marrow failure. The expression level of HDAC1 has been shown to be involved in the cell fate of hematopoietic progenitors. Overexpression of HDAC1 in hematopoietic progenitor cells transplanted into mice favors erythro-megakaryocytic differentiation at the expense of myeloid differentiation. Conversely, HDAC1 knockdown is mediated by small interfering RNA increases in myeloid differentiation and disrupts the erythroid differentiation of progenitor cells. The expression profile of HDAC1 is regulated by hematopoietic transcription factors. Thus, C/EBPs inhibits HDAC1 transcription in normal myeloid differentiation, while GATA-1 activates it during erythroid-megakaryocytic differentiation [76].

HDAC3 is another class I HDAC which was found to negatively regulate human HSC expansion. Specific HDAC3 knockdown significantly enhances CD34+ cell expansion without affecting cell differentiation in vitro [77]. Furthermore, HDAC3 can associate with the transcription factor GATA2 in the nucleus of the leukemic cell line KG-1 and in HEK cells, thereby repressing its transcriptional activity [78]. In mice, HDAC3 is essential to produce the earliest lymphoid progenitor cells and for HSC self-renewal in mice [79]. In zebrafish embryos, the emergence of HSCs from the ventral wall of the dorsal aorta is controlled by the nuclear receptor co-repressor 2 (NCOR2) [80]. During this process, NCOR2 is crucial for HSC development by cooperating with HDAC3 to repress FOS transcription. Knockdown of NCOR2 upregulates FOS, which induces VEGFD expression and the subsequent enhancement of Notch signaling, leading to the repression of hemogenic endothelial specification, a process required for HSC generation [81].

HDAC8, another member of the class I family, is involved in the maintenance of long-term hematopoietic repopulation. This enzyme interacts with p53 and modulates p53 activity by deacetylation. When exposed to genotoxic and hematopoietic stress, HDAC8-deficient long-term hematopoietic stem cells exhibit increased apoptosis associated with p53 hyperactivation [82].

Class II and class III HDACs are important players in HSC homeostasis and aging. A member of HDAC class IIa, HDAC5, controls HSC homing by downregulating the membrane receptor CXCR4 (chemokine C-X-C receptor-4) transcription though deacetylation of p65, a subunit of NF-κB. The interaction between CXCR4 and SDF-1 is an essential hub of the homing process [83]. Upon inhibition of HDAC5, the acetylated protein p65 binds to the promoter site of CXCR4, which leads to an increase in transcription and membrane expression, and the subsequent enhancement of SDF-1/CXCR4-mediated homing and engraftment of human HSCs [84].

Sirtuins, members of the class III family, protect HSCs against aging [85]. SIRT1 is essential for the homeostatic maintenance of the HSC pool as it promotes the activation and nuclear localization of its substrate, FOXO3. Young HSCs lacking SIRT1 share several characteristics with aged HSCs, such as the accumulation of damaged DNA and a similar expression profile. This suggests that SIRT1 could protect HSC from aging. Moreover, SIRT1 is involved in the regulation of lineage specification. Indeed, its loss induces anemia as well as a substantial expansion of the myeloid compartment, in particular granulocyte-monocyte progenitors (GMP), to the detriment of the lymphoid compartment [86]. SIRT3 is another key regulator of physiological aging of HSCs, reducing oxidative stress by modulating the acetylation level of mitochondrial proteins. Its expression decreases with age, contributing to the increase in ROS levels, and thus to the deterioration of the function of aged HSC. Upregulation of SIRT3 in aged HSCs improves their regenerative capacity [87]. SIRT6 also plays a role in HSC homeostasis. Its deletion results in an aberrant stimulation of the WNT signaling which leads to an abnormal proliferation of HSCs. Mechanistically, SIRT6 interacts with the transcription factor LEF1 and deacetylates H3K56ac, thereby repressing the transcription of WNT target genes. The pharmacological inhibition of the WNT pathway corrects the aberrant proliferation and malfunction of HSC-lacking SIRT6 [88]. Also, SIRT7 enhances the potential of aged HSCs to regenerate, by directing a mitochondrial unfolded protein response (UPRmt) regulatory branch. It has been suggested that the transcription factor NFR1 enhances SIRT7 recruitment to mitochondrial ribosomal proteins (mRPs) and mitochondrial translation factors (mTFs) promoters, repressing their expression and subsequently mitochondrial activity and proliferation. Inactivation of SIRT7 in HSCs leads to a decrease in quiescence, a rise in mitochondrial protein folding stress (PFSmt), and a disruption of their regenerative capacity. Aged HSCs are characterized by reduced SIRT7 expression. Upregulation of SIRT7 improves their regenerative capacity [89].

### 4.2. Role of HDACs during Erythropoiesis

Experiments conducted both in vivo and in vitro suggest various roles for HAT/HDACs in the erythroid lineage. For example, the HDACi FK228 and TSA-enhanced erythroid cell production from CD34^+^ cells exposed to IL-3, abrogated it in the presence of EPO. Furthermore, FK228 induces the apoptosis of CD36^+^GPA^low/−^ and CD36^+^GPA^high^ erythroid cells under EPO exposure [90]. Since these first observations, the underlying mechanisms and the specificity of action of HDACs on erythropoiesis are being progressively deciphered, and are reviewed below.

#### 4.2.1. HDACs and the Regulation of γ-Globin Gene Expression

Class I HDACs are able to interact with specific erythroid lineage-transcription factors, thus playing a role in the regulation of the human γ-globin (huγ) gene. For example, HDAC1 contributes to the Ikaros-mediated repression of huγ during the transition from γ- to β-globin in ontogenic development. Specifically, Ikaros promotes the recruitment of a complex with repressosome activity, which includes HDAC1, GATA1, FOG1, as well as components of the Mi-2/NuRD complex at huγ promoters. The formation of this repressosome decreases the frequency of remote chromatin interactions between the huγ promoters and the locus control region (βLCR). This process thus ultimately leads to specific silencing of the huγ gene [91]. HDAC1 is also able to interact with and repress NF-E4, a transcription factor involved in the expression of huγ genes. NF-E4 acetylation at lysine residue 43 (K43) releases it from its interaction with HDAC1, potentially increasing its ability to activate at the huγ promoter [92]. 

By associating with NCOR1 (nuclear receptor co-repressor 1), HDAC3 forms a protein complex which occupies the γ-globin promoter, playing a role in the developmental silencing of fetal globin genes. In contrast, the displacement of the HDAC3/NCOR1 complex mediated by short-chain fatty acids (SFCA) promotes the recruitment of RNA polymerase II associated with an enhancement in histone H3 and H4 acetylation at the huγ gene promoter. Moreover, in human erythroid progenitors, specific siRNA knockdown of endogenous HDAC3 results in an increased transcription at the huγ gene promoter [93]. In addition to its role in huγ gene regulation, it is noteworthy that HDAC3 can bind to the promoter of the hepcidin gene, a key hormone in the regulation of iron metabolism. Therefore, the HDAC3/NCOR1 complex may be involved in the downregulation of hepcidin. In iron-deficient mice, pharmacological inhibition of HDAC3 upregulates hepcidin expression. Furthermore, HDAC3 knockdown neutralizes the suppression of hepcidin induced either by erythroferrone or by the inhibition of bone morphogenetic protein signaling [94].

Unlike HDACs 1 and 3, HDAC9 upregulates γ-globin gene expression. Indeed, chromatin immunoprecipitation (ChIP) assays have shown that HDAC9 binds in vivo in the region upstream of the Gγ-globin gene promoter. Furthermore, small interfering RNA (siRNA) knockdown of HDAC9 in primary human erythroid progenitors results in γ-globin gene silencing. Conversely, the forced expression of HDAC9 simultaneously increases γ-globin and HbF mRNA levels. Because multiple transcription factor binding regions for myocyte enhancer factor 2 (MEF2) have been identified in the γ-globin promoters of the erythroleukemic line K562, it has been suggested that MEF2 may be involved in the recruitment of HDAC9 to these promoters [95].

#### 4.2.2. Class I HDACs in Terminal Erythroid Differentiation

The KO of HDAC1/2 in hematopoietic cells is lethal in mice, resulting in severe hematopoietic defects predominant in the erythroid lineage, with a high apoptotic rate [96]. HDAC1/2 are involved in erythropoiesis at several levels. First, they are integrated into complexes involved in terminal erythroid differentiation, including SIN3A NuRD or CoREST [97,98].

(i) The Sin3A/HDAC1 corepressor complex can interact with EKLF, an essential transcription factor in the activation of the β-globin gene [99]. The CBP/p300-mediated acetylation of EKLF promotes its interaction with the SWI/SNF chromatin remodeling complex, and consequently, transcriptional activation at the β-globin gene promoter [100]. Conversely, repression of EKLF could involve its deacetylation by HDAC1 [99]. In murine erythroleukemia (MEL) cells, PU.1-associated MeCP2 mediates the recruitment of mSin3A/HDAC1 to the IVS2 (intervening sequence 2) region binding site located in the β-globin gene intron, repressing its expression. During MEL differentiation, the PU.1-MeCP2-mSin3A/HDAC complex breaks away from this region, allowing β-globin expression [101]. 

(ⅱ) HDAC1 acetylation status also affects the activity of the HDAC1-containing NuRD complex. NuRD is essential for GATA1-mediated gene activation/repression processes. During the early proliferative phase of erythropoiesis, NuRD deacetylates histones via HDAC1 and rearranges chromatin to repress the expression of GATA1 target genes. On the contrary, during GATA1-directed erythroid terminal differentiation, p300/CBP-mediated acetylation inhibits HDAC1 deacetylase activity within the NuRD complex, which then exerts an activating effect on gene expression [102].

(ⅲ) HDAC1 and HDAC2 are also components of a repressor complex that comprises CoREST, the histone demethylase LSD1, and the transcriptional repressors Gfi-1 and Gfi-1b. Gfi-1b, by associating with CoREST-LDS1, recruits these cofactors to the target promoters. Inhibition of CoREST and LSD1 disrupts the differentiation of erythroid, megakaryocytic, and granulocytic cells, as well as primary erythroid progenitors [103].

Another mechanism of the HDAC1-mediated control of erythropoiesis is based directly on GATA1 deacetylation. Indeed, HDAC1 and CBP are required for controlling GATA1 acetylation levels, which regulate its transcriptional activity and erythroid differentiation [104]. In MEL cells, the acetyltransferase, CBP, strongly increases the transcriptional activity of GATA1 by binding to its zinc finger domain [105]. in G1E cells, two motifs rich in acetylable lysine residues located at the C-terminal ends of zinc fingers are particularly important, as their mutations abolish GATA1 function [106]. Moreover, acetylation of GATA1 enables binding between GATA1 and BRD3, a member of the BET protein family, at major erythroid promoters. The abolition of GATA1/BRD3 interaction by pharmacological inhibition disrupts GATA1-dependent erythroid differentiation [107].

HDAC1 is also involved in the expression of PU.1, a negative regulator of erythroid commitment. Mechanistically, HDAC1 deacetylates TAF9, a member of the TFIID complex, allowing TAF9 to fix the PU.1 gene promoter and activate its expression. During erythropoiesis, acetylation of HDAC1 occurs, which reduces its deacetylase activity and thus promotes acetylation of TAF9. Acetylation of TAF9 disrupts its binding to the PU-1 gene promoter, and disassembles the TFIID complex, resulting in the repression of PUI transcription and the erythroid commitment of multipotent myeloid progenitors [108,109,110]. 

Finally, in mice fetal liver cells, HDAC2 knockdown did not affect erythroid cell proliferation, differentiation, or apoptosis, but had a negative effect on nuclear condensation and enucleation, showing that Class I HDACs may have a different level of action on erythroid differentiation, depending on the stages and the nature of erythroid cells [111].

#### 4.2.3. HDAC Class II during Erythroid Differentiation

Among the class II HDACs, two enzymes appear to play a particularly important role during erythropoiesis: HDAC5 and HDAC6. HDAC5, a member of HDAC class IIa, plays both a negative and a positive role during erythropoiesis, depending on the cell system and the stage of differentiation. In MEL cells, HDAC5, colocalized with GATA1 and their co expression, repressed the GATA1 transcriptional activity, whereas during MEL differentiation, the complex dissociates and HDAC5 partially shuttles to the cytoplasm [112]. In erythroid cells, EPO stimulation activates the protein kinase D, which by presumably phosphorylating HDAC5, promotes dissociation of the HDAC5/GATA1 complex, leading to acetylation of the transcription factor GATA1, a process essential for erythropoiesis [106,113]. HDAC5 knockdown increases the response to EPO, while disruption of PKD signaling impairs erythroid differentiation under EPO exposure. HDAC5 deficiency in a mouse model results in resistance to anemia, increased erythroid commitment of progenitors, and promotes EPO-independent erythroid maturation [113].

HDAC5 is also involved in the regulation of erythropoiesis through the nuclear remodeling shuttle erythroid (NuRSERY) complex which contains HDAC5, GATA1, EKLF and pERK. This complex allows GATA1 and EKLF transportation from the cytoplasm to the nucleus. ERK phosphorylation is thought to be required for the formation of NuRSERY. Since pERK decreases during erythroid differentiation, the NurSERY complex seems to be predominantly involved in the early steps of erythropoiesis [114].

More recently, Wang et al. showed that HDAC5 expression increased significantly in the later phase of human erythropoiesis and that HDAC5-mediated deacetylation of histone and non-histone proteins was required for the survival, proliferation and enucleation of erythroblasts. Indeed, the lack of HDAC5 resulted in increased apoptosis and impaired enucleation of erythroblasts, and led to the acetylation and activation of the pro-apoptotic molecule p53. Furthermore, in late-stage erythroblasts, HDAC5 deficiency or pharmacological inhibition increased H4 (K12) acetylation, together with decreased chromatin condensation, arguing for an HDAC5 role in chromatin condensation during erythropoiesis [115].

HDAC6, belonging to HDAC class IIb, is another deacetylase involved in the enucleation process of terminal erythroid differentiation. Indeed, in mouse fetal erythroblasts, mDia2, a formin family effector protein, is an important player in enucleation because it promotes the establishment of the actomyosin contractile ring (CAR). The acetylation status of lysine, located in the homology 2 domain of formin, is involved in the regulation of mDia. Indeed, its deacetylation by HDAC6 in mouse erythroblasts leads to mDia2 activation, followed by CAR formation at the cleavage furrow and enucleation [116]. Inactivation of HDAC6 results in the accumulation of acetylated mDia2, which disrupts CAR establishment and ensuing cytokinesis and enucleation processes. Overexpression of non-deacetylated mDia2 corrects the enucleation defect associated with HDAC6 invalidation [116]. However, it should be noted that mDia2-deficient erythroblasts can retain their enucleation capacity, suggesting that other mechanisms are involved in this process [117]. In humans, clinical studies using HDAC6 inhibitors, such as ricolinostat (ACY-1215), used as monotherapy or in combination with other chemotherapeutic molecules, revealed the frequent occurrence of anemia as a secondary effect [118,119,120,121,122,123,124,125]. HDAC6 is expressed early during human erythropoiesis. ACY-1215 and HDAC6 knockdown in in vitro erythroid cultures from human CD34+ cells induced a blockage at the transition from CFU-E/Pro-E to later precursor stages, and a decreased JAK2/STAT5 response to EPO stimulation [126]. The underlying mechanism involved the protein 14-3-3ζ as a direct target of HDAC6 in human erythropoiesis. 14-3-3ζ is a crucial player involved in hematopoiesis by different mechanisms, depending on the cell lineage. In murine hematopoietic stem cells, 14-3-3ζ interacts with LNK, preventing it from binding to, and inhibiting, JAK2. LNK-dependent negative control of JAK2 is crucial, as evidenced by the more rapid development of myeloproliferative neoplasm in LNK^−/−^ mice expressing mutated JAK2^V617F^, and the description of myeloproliferative neoplasms associated with LNK mutations in humans [127,128]. In erythroid cells, the ability of LNK to interact with JAK2 is controlled by the level of acetylation of 14-3-3ζ, which is regulated by HDAC6. HDAC6 inhibition, by increasing the 14-3-3ζ acetylation level, decreased its interaction with LNK, allowing the latter to interact and inhibit JAK2 signaling in response to EPO [126]. Consequently, pharmacological inhibition of HDAC6 could represent an interesting therapeutic strategy in the treatment of myeloproliferative neoplasms, such as polycythemia vera (PV), characterized by the activating mutation JAK2^V617F^. However, the relevance of HDAC6 control of erythropoiesis via the level of 14-3-3ζ acetylation determining LNK-JAK2 interaction in the context of JAK2^V617F^ mutation remains to be demonstrated. Indeed, recent data have shown that in leukemic cells carrying the JAK2^V617F^ mutation and in murine models of myeloproliferative neoplasms induced by the MPL^W515L^ mutation, HDAC11 more than HDAC6 was required for cell proliferation and survival via the control of the JAK-STAT pathway [129].

## 5. Conclusions

HDACs are enzymes capable of deacetylating a wide variety of functional and structural, nuclear and cytoplasmic proteins. They are amongst the promising therapeutic targets in cancer therapy and have led to the development of numerous HDACi. Despite the clinical interest in these molecules, they have side-effects, notably hematological ones, due to the wide range of functions that HDACs exert on hematopoiesis, including the regulation of HSC homeostasis and erythropoiesis. In the erythroid lineage, HDACs are involved at all steps, from the regulation of the erythro-megakaryocytic commitment to enucleation, and from the control of EPO signaling to the globin switch. This remarkable diversity of HDACs function during erythropoiesis occurs through multiple mechanisms, including chromatin accessibility, transcription factor activity or cytoplasmic targets regulating apoptosis, protein quality control pathways or signaling between membrane receptor activation and gene transcription. Thus, it seems crucial to pursue the effort to decipher the role of HDACs during normal, but also pathological erythropoiesis, such as polycythemia vera and myelodysplastic syndrome, keeping in mind that they remain potential selective targets in these diseases.

**Table 2 ijms-23-09790-t002:** Cellular functions of HDACs involved in hematopoiesis and associated potential substrates.

Member	Related Cellular Functions	Substrates
HDAC1	Positive role in HSC homeostasis through the SIN3A complex [130] Positive role in erythro-megakaryocytic differentiation at the expense of myeloid differentiation in mouse hematopoietic progenitors [76] Repression of EKLF via the Sin3A-HDAC1 complex. EKLF is a potential target that can be deacetylated by HDAC1 at residue K302 [99] Histone deacetylation and chromatin remodeling into a repressive structure via the HDAC1-NuRD complex. Acetylation of HDAC1 within NuRD by p300/CBP abolishes its deacetylase activity, allowing NuRD to activate genes during GATA1-directed erythroid differentiation [102] Differentiation of erythroid, megakaryocytic, and granulocytic lineages via the CoREST complex [131] Coactivator of PU.1 expression. HDAC1 deacetylates TAF9, allowing TAF9 to bind and activate the PU.1 gene promoter [108,109,110] Human γ-globin gene silencing via the NuRD repressor complex [91]	EKLF (K302) Histones H3 and H4 TAF9
HDAC2	Positive role in HSC homeostasis, through the SIN3A complex [130] Positive role in erythro-megakaryocytic differentiation at the expense of myeloid Differentiation of erythroid, megakaryocytic, and granulocytic lineages via the CoREST complex [131] Positive role in chromatin condensation and enucleation [111]	
HDAC3	Negative regulation of human HSC expansion [77] Production of the earliest lymphoid progenitors and self-renewal of HSCs in mice [79] Positive role in the specification of the hemogenic endothelium, a prerequisite for HSC emergence, through cooperation with NCOR2 in a manner which represses FOS, in zebrafish [81] Repression of GATA2 transcriptional activity on HSC survival and proliferation by direct interaction [78] Human γ-globin gene silencing via the NCOR1 complex. Displacement of HDAC3 from the promoter site results in increased acetylation of H3 and H4 [93]	Histones H3 and H4
HDAC5	Control of HSC homing by downregulation of CXCR4 membrane receptor transcription via deacetylation of p65, a subunit of NF-κB [83] Positive role in human erythroblast survival, proliferation, nuclear condensation, and enucleation. HDAC5 deficiency induces acetylation and activation of the pro-apoptotic molecule p53, but also acetylation of H4 (K12) associated with decreased chromatin condensation [115]	p65 p53 and histone H4
HDAC6	Positive role in human erythroid differentiation through modulation of JAK2 signaling Positive role in CAR formation, cytokinesis, and enucleation via deacetylation of mDia2 in mouse fetal erythroblasts [116] Actin filament assembly required for human platelet production via CTTN deacetylation [75]	14-3-3ζ mDia2 CTTN
HDAC8	Positive role in maintaining long-term hematopoietic repopulation through deacetylation of p53 in LT-HSC [82]	p53
HDAC9	Upregulation of human γ-globin genes [95]	
SIRT1	Positive role in the maintenance of HSC homeostasis, by promoting the localization and nuclear activation of its substrate FOXO3 [85] Regulation of lineage specification in HSCs [86]	FOXO3
SIRT3	Regulation of physiological aging of HSCs by reducing oxidative stress via modification of global mitochondrial protein acetylation [87]	SOD2
SIRT6	Key role in HSC homeostasis by repressing transcription of WNT target genes via interaction with transcription factor LEF1 and deacetylation of H3K56ac [88]	Histone H3 (K56)
SIRT7	Positive role in the regenerative capacity of aged HSCs by directing a regulatory branch of the mitochondrial unfolded protein response (UPRmt) [89]	

## Figures and Tables

**Table 1 ijms-23-09790-t001:** Classification of HDAC family enzymes.

HDAC Class	Human Protein	Cellular Localization	Cofactor
Class I	HDAC1	Nucleus	Zn^2+^
	HDAC2	Nucleus	
	HDAC3	Nucleus/Cytoplasm	
	HDAC8	Nucleus	
Class IIa	HDAC4	Nuc leus/Cytoplasm	Zn^2+^
	HDAC5	Nucleus/Cytoplasm	
	HDAC7	Nucleus/Cytoplasm	
	HDAC9	Nucleus/Cytoplasm	
Class IIb	HDAC6	Nucleus/Cytoplasm	Zn^2+^
	HDAC10	Nucleus/Cytoplasm	
Class III	SIRT1	Nucleus	NAD^+^
	SIRT2	Cytoplasm	
	SIRT3	Mitochondria	
	SIRT4	Mitochondria	
	SIRT5	Mitochondria	
	SIRT6	Nucleus	
	SIRT7	Nucleus	
Class IV	HDAC11	Nucleus/Cytoplasm	Zn^2+^

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
