# Peer review of "Histone Deacetylases Function in the Control of Early Hematopoiesis and Erythropoiesis"

_ijms, 2022, doi:10.3390/ijms23179790_

Round 1

Reviewer 1 Report

  1. In the Conclusion, the clarity of the presentation needs to be improved.

  1. A graphical abstract at the end of the Discussion may further make the paper more reader-friendly.

Author Response

I modified.

Reviewer 2 Report

The manuscript entitled “Histone desacetylases function in the control of early hematopoiesis and erythropoiesis” is well written, wonderfully drafted, and covered HDAC and hematopoiesis. However, it did not provide any basic idea about regulation of HDAC (at least a basic concept).

Author Response

I modified.

Reviewer 3 Report

The review on the histone deacetylase function in hematopoiesis is clearly described based on the available newest literature data. There are a new topic in hemathematologyresearch results that may contribute significantly to the development of diagnosis and therapies of the diseases related to hematopoiesis.

No improvements are suggested to present article version.

Author Response

I modified.